Horsenettle (Solanum carolinense) fruit bacterial communities are not variable across fine spatial scales

Heminger Ariel R. 1 2
Belden Lisa K. 2 3
Barney Jacob N. 1 2
Badgley Brian D. 1 2
Haak David C. dhaak@vt.edu 1 2
1 School of Plant and Environmental Sciences, Virginia Polytechnic Institute and State University (Virginia Tech) , Blacksburg , VA , United States of America
2 Global Change Center, Virginia Polytechnic Institute and State University (Virginia Tech) , Blacksburg , VA , United States of America
3 Department of Biological Sciences, Virginia Polytechnic Institute and State University (Virginia Tech) , Blacksburg , VA , United States of America
Czajkowski Robert
Electronic publication date: 2021 Nov 8
Publication date: 2021
Volume: 9
Electronic Location ID: e12359
Received 2021 Jun 14; Accepted 2021 Sep 30
Copyright: ©2021 Heminger et al.
Copyright year: 2021
Copyright holder: Heminger et al.
License: This is an open access article distributed under the terms of the Creative Commons Attribution License, which permits unrestricted use, distribution, reproduction and adaptation in any medium and for any purpose provided that it is properly attributed. For attribution, the original author(s), title, publication source (PeerJ) and either DOI or URL of the article must be cited.
License URL: https://creativecommons.org/licenses/by/4.0/

Keywords: Solanum carolinense, Bacteria, Microbiome, Fruit microbiome, Spatial distance

Funding: Global Change Center at Virginia Tech This project was supported by an award to David C. Haak, Lisa K. Belden, Jacob N. Barney, and Brian D. Badgley, from the Global Change Center at Virginia Tech. The funders had no role in study design, data collection and analysis, decision to publish, or preparation of the manuscript.

==============================
Fruit house microbial communities that are unique from the rest of the plant. While symbiotic microbial communities complete important functions for their hosts, the fruit microbiome is often understudied compared to other plant organs. Fruits are reproductive tissues that house, protect, and facilitate the dispersal of seeds, and thus they are directly tied to plant fitness. Fruit microbial communities may, therefore, also impact plant fitness. In this study, we assessed how bacterial communities associated with fruit of Solanum carolinense, a native herbaceous perennial weed, vary at fine spatial scales (<0.5 km). A majority of the studies conducted on plant microbial communities have been done at large spatial scales and have observed microbial community variation across these large spatial scales. However, both the environment and pollinators play a role in shaping plant microbial communities and likely have impacts on the plant microbiome at fine scales. We collected fruit samples from eight sampling locations, ranging from 2 to 450 m apart, and assessed the fruit bacterial communities using 16S rRNA gene amplicon sequencing. Overall, we found no differences in observed richness or microbial community composition among sampling locations. Bacterial community structure of fruits collected near one another were not more different than those that were farther apart at the scales we examined. These fine spatial scales are important to obligate out-crossing plant species such as S. carolinense because they are ecologically relevant to pollinators. Thus, our results could imply that pollinators serve to homogenize fruit bacterial communities across these smaller scales.

Introduction

Plants, like other organisms, host diverse communities of microorganisms that are typically dominated by bacteria (Stone, EA & Jackson, 2018). Bacteria associated with above-ground portions of plants (the phyllosphere) play an important role in host development and health (Gibbons & Gilbert, 2015; Grover et al., 2011; Richardson et al., 2009; Selosse, Baudoin & Vandenkoornhuyse, 2004). Thus, understanding how these bacterial communities are formed and the factors that shape changes in these communities can lead to important insights about plant health. This is especially true when we consider tissues like fruits, which are directly tied to plant fitness (Compant et al., 2011; Eck et al., 2019; Nelson, 2018; Tewksbury et al., 2008). Despite this importance, few studies have examined patterns of bacterial community diversity among fruits (Compant et al., 2011; Miura et al., 2017).

A first step in understanding what shapes plant-associated microbial communities is characterizing variation over time and space. The tendency for communities to become increasingly dissimilar with distance is often referred to as “distance decay” (Finkel et al., 2012; Morrison-Whittle & Goddard, 2015; Peay, Garbelotto & Bruns, 2010), which have been observed in microbial communities (Feng et al., 2019). Viewed through an evolutionary lens, this pattern of community turnover could arise from two distinct mechanisms: local adaptation and dispersal limitation (Bell, 2010). In the first case, communities change concordantly with niche turnover along an abiotic or biotic gradient. In the latter case, community change occurs in the absence of an underlying environmental or resource gradient, exhibiting geographical patterns (Barreto et al., 2014). Studies of phyllosphere microbes have found mixed evidence for these mechanisms. For instance, a recent study of Tamarix leaves found that bacterial community structure was influenced by both geographic distance and environmental heterogeneity (Finkel et al., 2012). Yet, it is challenging to find generalizable patterns, as most studies of phyllosphere communities focus on leaves over large spatial scales, maximizing the potential to detect community differences, even though not all studies find evidence of distance decay. For example, a study of Pinus spp. at distances ranging from 1,800–2,400 km found no evidence that microbial communities became more distinct with increasing distance (Redford et al., 2010). In contrast, a recent study of Magnolia grandiflora leaves at distances up to 452 m found that microbial communities did become more distinct the further away they were (Stone & Jackson, 2016).

While widespread, bacteria may not be evenly distributed over large spatial scales (Bahram et al., 2018; Delgado-Baquerizo et al., 2018). Microbial communities on plants can be regulated by local biotic and abiotic conditions, including plant host species, geographic location, light, temperature, wind, and humidity (Berg et al., 2014; De Vries et al., 2012). In addition, the factors that shape bacterial distribution at global scales, such as soil pH and temperature (Bahram et al., 2018; Delgado-Baquerizo et al., 2018), may be more nuanced or attenuated at local levels (Igwe & Vannette, 2019; Storey et al., 2018). Further, the factors shaping microbial diversity may interact across scales to influence community assembly within particular communities, e.g., rhizosphere versus leaves (Levin, 1992). For example, using genetic mutants, Lebeis et al. (2012) demonstrated that Arabidopsis plants are capable of structuring soil bacterial communities. In contrast, Ottesen et al. (2016) found that in the tomato phyllosphere when host and other biotic factors were ruled out using plastic plants, bacterial communities were similar between the plastic plants and normal controls.

Studies at finer spatial scales are more limited (Stone & Jackson, 2016), and even fewer studies have been conducted in fruit microbial communities (Miura et al., 2017). Fruit microbial communities are likely shaped by a combination of biotic and abiotic factors similar to those that shape the leaf microbial communities. Yet, recent studies have also shown that fruit and flower microbial communities differ from other above ground plant tissues (Ottesen et al., 2013; Rasmann et al., 2012; Shade, McManus & Handelsman, 2013). One possible explanation for this is that these tissues interface directly with pollinators, biotic sources of additional microbes (Russell et al., 2019), especially in obligate outcrossing species (Wei & Ashman, 2018). Since pollinators transfer microbes from one flower to the next (Ushio et al., 2015), they may contribute to homogenizing microbial communities locally at these finer spatial scales.

Fruit and flowers are especially warranted to studies at finer spatial scales, as pollinator interactions are most likely to impact these tissues (Rebolleda-Gómez et al., 2019; Ushio et al., 2015; Zemenick, Vannette & Rosenheim, 2021). Recent studies found that plant communities with a similar floral visitor community have similar microbes (Rebolleda-Gómez & Ashman, 2019; Zemenick et al., 2019). However, they found that while flowers were hubs for arthropod visitation, flowers did not necessarily serve as hubs for microbes, which suggests a degree of host sorting (Zemenick, Vannette & Rosenheim, 2021) which supports the idea that floral visitors shape microbial communities. However, little is known regarding how this could impact fruit microbial communities.

Here, we hypothesized that bacterial communities associated with fruit that are spatially closer to one another will be more similar. This is because they are more likely to share a similar environment and be exposed to the same pollinator, since pollinators are known to visit multiple flowers. To address this hypothesis, we sampled the fruit of Carolina horsenettle, Solanum carolinense, a native herbaceous perennial weed that is an obligate outcrossing species, at a fine scale (under 450 m) and compared bacterial community composition at different distances to determine if we observed distance decay in the bacterial communities associated with fruit.

Materials & Methods

Study species: Solanum carolinense

Solanum carolinense L. (Solanaceae) is a native herbaceous perennial weed found throughout most of the eastern United States. This species reproduces both via underground horizontal roots (rhizomes) and by seed. Inflorescences are in clusters of 1–12 blossoms and are frequently perfect and functionally hermaphroditic. Solanum carolinense is known to be an obligate out-crosser, meaning that it cannot self-pollinate and relies on pollinators, specifically buzz pollinators, for fertilization (Hardin et al., 1972). Bombus impatiens, the eastern bumblebee, is a common pollinator of S. carolinense in the eastern United States. Fruits of S. carolinense are yellow to orange in color and 1–2.5 cm in diameter, and the reproductive season for S. carolinense is from late summer until the first frost (Bassett & Munro, 1986). S. carolinense was chosen for this study because it grows in a variety of habitats and can be found at relatively high abundances locally. S. carolinense is closely related to Solanum lycopersicum, so this work could indirectly provide insights into bacterial community similarities in commercially important Solanum species.

Sampling, processing, and DNA extraction

A total of 23 fruit samples, from nine different sampling locations of S. carolinense, were collected from around Blacksburg, VA, USA on November 15th, 2017 (Fig. 1, Table S1). The use of the site was approved by the Department of Parks and Recreation of the town of Blacksburg. Sample locations ranged in distance from 2 to 450 m apart, and the total number of individual plants sampled ranged from 2 to 5 individuals/site. Since, S. carolinense is a perennial species that can spread via underground rhizomes generating clusters of clonal plants it is possible that multiple of S. carolinense in a collection site are genetically identical. One fruit sample per S. carolinense ramet was collected aseptically in 50 ml falcon tubes and stored at −80 °C until DNA extraction. Prior to DNA extraction, samples were sliced using EtOH flame sterilized forceps and scalpels so that the sample included both surface and internal microbes. The samples were then disrupted with beads. DNA was extracted from samples using the DNeasy PowerSoil® Kit (Qiagen USA catalog no. 12888-100).

Figure 1 Sampling locations and number of plants collected from each location.

Sampling locations in Blacksburg, VA. Blue circles indicate Solanum carolinense locations sampled. Map on the lower left corner shows the sampling location, indicated in blue, in relation to the eastern United States. Map was created using Google Earth, Maps Data: Google, © 2021.

Library preparation

LNA PCR

Mitochondria and chloroplast from plant tissues pose a challenge when investigating plant microbial communities, as they share an evolutionary history with bacteria and contain 16S rRNA genes (Sakai & Ikenaga, 2013). Thus, many of the primers used to analyze bacterial community structure will also bind and amplify mitochondria and chloroplast DNA (Ghyselinck et al., 2013). To reduce the amount of mitochondrial and chloroplast amplification in our samples, we used an initial locked nucleic acid (LNA) PCR with the primers 63f-mod and 1492r (Ikenaga et al., 2016; Yu et al., 2016). The LNA mixture contained: 12.5 µL Premix Hot Start Accustart II Supermix, 1.0 µL 63f-mod primer (20 pmol/ µL), 1.0 µL 1492r (20 pmol/µL), 2.0 µL LNA-Mit63 (20 pmol/µL), 2.0 µL LNA-Mit1492 (20 pmol/µL), 2.0 µL LNA-Pla63c (20 pmol/µL), 2.0 LNA-Pla 1492b (20 pmol/µL), 1.5 µL sterile water, and 1.0 µL of DNA template (Table S2). LNA PCR was performed in sets of eight, which consisted of seven samples and one negative control without DNA template per run. The amplification conditions were 94 °C for 1 min to denature the DNA, followed by 30 cycles of 94  °C for 1 min, 70 °C for 1 min to anneal the LNA oligonucleotides, 54 °C for 1 min to anneal primers, and 72 °C for 2 min, and a final extension step of 72 °C for 10 min. PCR products and negative controls were visualized using a 1.5% agarose gel to ensure that samples had amplified correctly.

Illumina 16S PCR

We then amplified the V4 region of the 16S rRNA gene, using primers 515f and an individually barcoded 806R, with the LNA product as the template. PCR was run in duplicate (two 25 µL reactions) along with a negative control that did not contain any template DNA. The PCR mixture contained: 12 µL UltraClean PCR grade H2O, 10 µL of 5 Prime Hot Master Mix, 0.5 µL of forward primer 515f and 0.5 ul of reverse primer 806R (barcoded), and 2 µL of template DNA (Table S2). The amplification conditions were: 94 °C for 3 min for the initial denaturation, followed by 30 cycles of 94 °C for 45 s, 50 °C for 1 min, and 72 °C for 90 s, and a final extension of 72 °C for 10 min. Following PCR, the duplicate samples were combined and were run on an 1.5% agarose gel to ensure that samples had amplified correctly. Concentrations for the samples were recorded using a Qubit Fluorometer 2.0 with the HS DNA kit. Equimolar amounts of each sample were then pooled and cleaned using the Qiaquick PCR purification kit. Amplicon sequencing was completed at the Dana Farber Cancer Institute at Harvard University using 250 bp single end reads on the Illumina Mi-Seq.

Identifying ASVs

Demultiplexed raw reads were processed in the R computing environment (R Core Team 2020). The package manager Bioconductor Martin (2018) was used to implement packages, including DADA2 1.12.1 Martin (2018), ggplot2 3.2.1 (Wickham, 2015), DESeq2 1.24.0 (Love, Huber & Anders, 2014), phyloseq 1.28.0 (Mcmurdie & Holmes, 2013), and vegan 2.5-6 (Oksanen et al., 2019). Raw reads were quality processed using methods laid out in the DADA2 tutorial Martin (2018). The reads were first visualized using the “plotQualityProfile” function in DADA2, which allowed us to monitor for low quality reads. No trimming was necessary, so we maintained the full 250bp of the reads for analyses. The “learnErrors” function was then used to evaluate the error rate of the data set. The “derepFastq” function was used to combine identical sequences into unique sequences, and from this a table of amplicon sequence variants (ASVs) was generated using the “makeSequenceTable” function.

Chimeric reads were removed using the “removeBimeraDenovo” function, which identified 1,700 chimeras out of 4,225 input ASVs (Table S3). Taxonomy was assigned to the remaining ASVs using the “assignTaxonomy” function in DADA2 and the SILVA v123 database (Quast et al., 2013). Samples that could not be assigned to bacterial phyla were removed, which resulted in a total of 15 phyla remaining. Any mitochondria or chloroplast reads that persisted, along with unclassified reads, were also removed. After processing and quality control, our dataset consisted of 1874 ASVs across all 23 samples. From this final ASV set, phyla with read counts <10 were removed (Mcmurdie & Holmes, 2013). After this step, our dataset consisted of 1846 ASVs across all 23 samples including the following phyla: Acidobacteria, Actinobacteria, Armatiomonadetes, Bacteroidetes, Firmicutes, and Proteobacteria.

Richness and beta diversity

Comparisons of ASVs among each sampling location were conducted using observed richness and Shannon alpha diversity indices using the “plot_richness” function and visualized using phyloseq in R. Observed richness indicates the number of ASVs and provides information on the number of ASVs that are found within each sample per site. Shannon’s diversity index considers both ASV richness and evenness. Faith’s phylogenetic diversity was calculated using the “estimate_pd” function in btools (Battaglia, 2020). A Kruskal–Wallis rank sum test was used on both observed richness and Faith’s phylogenetic diversity to analyze differences between each of the sites.

To assess differences in bacterial community composition among locations, we compared beta diversity using two distance metrics: Bray–Curtis dissimilarity and Jaccard index. Bray–Curtis dissimilarity is based on ASV relative abundance, while Jaccard is based only on presence/absence of ASVs. Differences in Bray–Curtis dissimilarity and Jaccard were analyzed using PERMANOVA and were visualized using PCoA plots.

To test for spatial patterns of dissimilarity among locations, we conducted a Mantel test to estimate the correlation between two matrices. We chose to use the Mantel test because it allows us to compare spatial relationships with bacterial community composition and is commonly used in ecology. We used the Mantel test to look for correlations between the Bray–Curtis dissimilarity matrix and the spatial distance matrix among all 23 samples, where samples within each site were considered to have a distance of 0 m. Pearson’s product-moment correlation was used to test for differences. One of the assumptions of PERMANOVA is that sample size is equal with similar dispersion between samples. Since we knew that we had uneven sampling we tested the heterogeneous variance among microbial communities. This was tested using the function “betadisper”, which is a multivariate analog of Levene’s test for homogeneity of variances.

To look for core bacterial communities, we assessed which ASVs were most prevalent and their relative abundances. We conducted this using the “core_members” function from the microbiome package Battaglia (2020). We set our parameters such that ASVs needed to be present in at least 60% of the samples and had a relative abundance of at least 0.10%.

Results

To describe the composition of the bacterial communities associated with fruit tissue from each sampling location, we estimated relative abundances of different bacterial orders (Fig. 2). The orders Enterobacteriales, Sphingomonadales, Sphingobacteriales, Rhizobiales, Cytophagales, and Micrococcales were found to make up the greatest proportion of the relative abundance found at each sampling site (Fig. 2). Interestingly, some differences in abundance were found across sites. The orders Micrococcales, Caulobacterales, and Betaproteobacteriales were present in some sites and not found at others. Micrococcales was found at sites 2, 4, and 5, and Caulobacterales was found at sites 2, 5, and 7. Both Micrococcales and Caulobacterales were found at relatively low abundances compared to other prominent taxa. Betaproteobacteriales was found at relatively low abundances at all the sites except site 1. Flavobacteriales was also found at relatively low abundance but found at all the sites sampled. The most common bacterial genera by relative abundance were: Aureimonas, Chryseobacterium, Dyadobacter, Hymenobacter, Methlobacterium, Mucilaginibacter, Neorhizobium, Pantoea, Pedobacter, Pseudomonas, Rosenbergiella, Sphingomonas, Spirosoma, and Tatumella; these all had a relative abundance of ≥ 0.1%.

Figure 2 Relative abundance of microbial communities by sampling location.

Relative abundance of bacterial orders that constituted >2% of S. carolinense fruit microbiome, by sample location.

Bacterial community analysis

Shannon diversity, observed richness, and Faith’s phylogenetic diversity were then calculated for each sample location using abundance data to determine richness and evenness (Fig. 3). Although there was some separation of mean values in alpha diversity across sites, none of these differences were significant for observed richness (p = 0.2173, DF = 7, chi-squared = 9.5226) or Faith’s phylogenetic diversity via Kruskal–Wallis rank sum test (p = 0.1879, DF = 7, chi-squared = 10.012). The high values observed in mean alpha diversity likely result from high variation within sites (Fig. 1).

Figure 3 Alpha diversity of microbial communities associated with S. carolinense fruit sample locations.

Observed richness, Shannon, and Faith’s phylogenetic diversity by sampling locations.

Similarly, Bray–Curtis dissimilarity of S. carolinense fruit did not reveal any differences in bacterial community structure across sampling locations (PERMANOVA, p = 0.711, DF = 22, R2 = 0.3033). PCoA analysis revealed that the first two components of variation explained a combined 29.1% (18.3% and 10.8% respectively) of the variation in bacterial community composition, however we did not find any significant clustering of samples by site (Fig. 4). For example, samples from the same location (color in Fig. 4) tended to be dispersed across the axes of variation. Likewise, no differences in bacterial community composition among sampling locations was identified using the Jaccard index (Figure S1, PERMANOVA, p = 0.688, R2 = 0.30896). These findings were supported by a Mantel test of Bray–Curtis dissimilarity index and calculated spatial distances that failed to detect a relationship between bacterial diversity and distance (Fig. 5 Pearson’s, Mantel statistic r: 0.03343, p = 0.302, DF = 22). Together, these results suggest that fruit bacterial diversity is not differentiated at this spatial scale.

Figure 4 Beta diversity (Bray–Curtis) of sample locations.

PCoA results illustrating distances among microbial communities associated with S. carolinense fruit, colored by sample locations (Bray–Curtis, PERMANOVA, p = 0.711, DF = 22, R2 = 0.3033). The axes indicate the percentage of variation in the data with axis 1 (the first component) representing 18.3% of the variation and axis 2 (the second component) representing 10.8% of the variation.

Figure 5 Mantel test.

No observable relationship between Bray–Curtis Dissimilarly and spatial distance (geographic distance Euclidean) was detected between bacterial diversity and distance (r: 0.03343, p = 0.302).

To investigate the proportion of shared taxa among samples from all sites, we conducted an analysis of the core microbial community and found that just 25 ASVs were present in at least 60% of the samples with a relative abundance > 0.10% (Fig. 6). The top five taxa that dominated the core microbial communities, as determined by a relative abundance of > 1.0% and prevalence > 50% were Aureimonas spp., Pantoea spp., Sphingomonas spp., Hymenobacter spp., and Pedobacter spp. Thus, 0.0135% of ASVs we identified in our analysis were found in the majority of fruit and less than half of these were found with a relative abundance > 0.30%. Finally, we found that Aureimonas spp. persisted with a very high prevalence >90% at relative abundance levels up to 1.0%.

Figure 6 Core microbial communities.

Microbial genera that were present (prevalence) in at least 60% of the samples at a relative abundance of at least 0.10%.

Discussion

The advent of non-culture based microbial community profiling has led to a wealth of information about patterns and scales of community diversity among bacteria that form close associations with plant organs (Compant et al., 2011; Junker et al., 2011; Ottesen et al., 2016; Shade, McManus & Handelsman, 2013). Non-pathogenic fruit-associated bacteria have received less attention, yet these communities have important implications for host fitness. This study provides a first glimpse into the composition of fruit bacterial communities of S. carolinense. This study also is the first to explore patterns of bacterial community structure of fruit in natural populations at fine spatial scales. This is one of the first studies to be conducted in fruit at such a fine spatial scale. We found that bacterial community richness and diversity are similar at distances up to 450m, and there is no correlation between community structure and distance, suggesting no evidence of distance-decay at these fine spatial scales. Factors such as microenvironment or biotic interactions (pollinators, seed predators, etc.) may, therefore, be more important for shaping patterns of bacterial diversity among fruit. It is also possible that a lack of genetic diversity among host plants (S. carolinense can and does reproduce clonally through rhizomes) could account for a lack of distance decay. However, because rhizome growth is generally limited to about 1.25 m (Kiltz, 1930) from the ramet, we would expect to reduce variation at the site level (Imaizumi et al., 2006), which was not apparent in this study. Still, the role of host-genotype level filtering in shaping fruit microbial communities in this system remains an open question.

As with other studies, we found that fruits are colonized by a diverse array of bacteria. In our study, we found that the Enterobacteriales, Rhizobiales, and Cytophagales orders comprised about 63% of the relative abundance across all samples. Within these orders, the most frequently identified genera were Pantoea, Aurantimonas, and Methylobacterium. In other plant hosts, different genera seem to dominate. For instance, fruits of congeneric tomatoes (Solanum lycopersicum) are dominated by the genera Xanthomonas, Rhizobium, and Pseudomonas (Ottesen et al., 2013); the only overlapping genus identified in our study was Sphingomonas. These findings reinforce the importance of host species (Igwe & Vannette, 2019; Knief et al., 2010; Ottesen et al., 2016; Wei & Ashman, 2018) and environment (Knief et al., 2010) in shaping bacterial communities in fruit. Despite the substantial variation in diversity, we identified 25 ASVs that were present in a majority of samples (>60%) (Fig. 6). Subsetting these further we identified five taxa that were present in nearly every sample with a relative abundance > 0.10%, Aureimonas spp., Pantoea spp, Sphingomonas spp., Hymenobacter spp., and Pedobacter spp. These top five taxa have all previously been reported from different plant tissue samples. Aureimonas spp. have been identified in the phyllosphere of Galium album (Aydogan et al., 2016), the bark of Populus spp. (Li et al., 2018), and, recently in Actinidia deliciosa (Ares et al., 2021). Pantoea spp. have been identified as endophytes in the stems of tomatoes, Solanum lycopersicum (Dong et al., 2019). Pantoea spp. can also cause internal fruit rot in Cucurbitaceae (Kido et al., 2008). Sphingomonas spp. have been found in tomato specifically on the lower stems and leaves (Ottesen et al., 2013), can cause disease in Cucurbitaceae (Buonaurio, Stravato & Cappelli, 2001), and can be endophytes that produce gibberellins (Latif Khan et al., 2014). Hymenbobacter spp. have been observed on Hedera spp. (Smets et al., 2016) and in the phyllosphere of Galium album (Aydogan et al., 2016). Pedobacter spp. have been observed in the phyllosphere of Solanum tuberosum, which is closely related to our study species (Manter et al., 2010), as well as the leaves of Arabidopsis thaliana (Qi et al., 2021).

Comparing bacterial community diversity, we found a great deal of variation within and among sites (Figs. 3 and 4). This variation was not partitioned by site or distance (Fig. 5) and is consistent with diverse source pools and environmental filtering (Rebolleda-Gómez & Ashman, 2019). It is important, however, to consider these findings with the caveat that reduced sample sizes negatively impact alpha diversity statistics (Willis, 2019). Thus, while widespread, bacterial taxa may not be evenly distributed over large spatial scales (Bahram et al., 2018; Delgado-Baquerizo et al., 2018).

Environment is an important source and factor shaping leaf microbial communities across fine spatial scales. This has been observed by Stone & Jackson (2016) found that bacterial communities associated with the leaves of Magnolia grandiflora exhibited a distance-decay relationship across a similar distance to the present study (1–452 m). Ultimately, Stone & Jackson (2016) conclude that subtle differences in environmental conditions contributed to differences in microbial communities. The lack of distance-decay in the present study may be for several reasons. Our sampling may not have covered a sufficient range nor elevation (17 m) in environmental variation however we did have variation in slope and aspect (north and west facing) (Fig. 1).

Host species and genotype may also play a role in shaping microbial communities associated with the plant (Laforest-Lapointe, Messier & Kembel, 2016; Redford et al., 2010). However, there has been contrasting evidence in Solanum lycopersicum leaves, suggesting that the host plant is not different from a controlled plastic plant (Ottesen et al., 2016). In addition, plant organs may also play a role in the interactions shaping distance decay. Several studies have documented distinct microbial communities across plant tissues (Compant et al., 2011; Junker & Keller, 2015; Ottesen et al., 2013; Shade, McManus & Handelsman, 2013), including different tissues within the same organ (Hayes et al., 2021). This also means that microbial communities associated with particular plant organs may have a different set of organizing rules (Rebolleda-Gómez & Ashman, 2019; Wei & Ashman, 2018; Zheng & Gong, 2019). For instance, flowers and fruits that rely on pollinators, which spread microbes, may not display distance decay at finer spatial scales. Another important consideration is the production of secondary compounds within the fruit. As with other members of the Solanaceae S. carolinense fruit contain steroidal glycoalkaloids that likely impact microbial communities (Milner et al., 2011). Future studies can determine the relative impact of SGAs on microbial communities by isolating the endophytic component of fruit microbial communities. By understanding how and when microbial communities are shaped in fruits and flowers we can learn how this impacts host fitness (Baltrus, 2020; Taylor et al., 2014).

Dispersal is another important component acting across these smaller spatial scales. For instance, Miura et al. (2017) investigated leaves and fruits in vineyards in Chile that were within 35 km of one another and found that at spatial scales <2 km fungal community dissimilarity increased with distance; however, the bacterial communities did not show similar patterns. This may be because fungal spores tend to have more limited dispersal than bacteria (Peay & Bruns, 2014). However, over large distances (1,800–2,400 km) Pinus spp. leaves showed no variation with an increase in distance (Redford et al., 2010), yet there was a distance-decay effect observed in Tamarix spp. (Finkel et al., 2012). We are still in the early phases of understanding how microbial communities are shaped across the landscape, particularly in reproductive tissues. Biotic interactions can drive facilitated bacterial dispersal (Russell et al., 2019). As an obligate outcrossing species, S. carolinense depends on pollinators. Pollinators leave microbial footprints on the flowers that they visit (Ushio et al., 2015), which may influence floral microbial communities, and possibly fruit microbial communities. Thus, the observed similarities among bacterial communities in our study could result from pollinator transmission, though we do not yet know if bacterial communities are transferred from flowers to fruit in this system. In addition to pollinators, seed feeding insects, such as the Lygaeidae, pierce fruit and feed on the developing seeds. This process introduces microbes to fruits (Tewksbury et al., 2008) and may be another source of bacterial dispersal. Indeed, our clustering analysis identified some axes of separation for samples within a site, which could be indicative of these sorts of biotic filters, yet more work needs to be done to test this hypothesis.

Conclusions

In our study, we found that richness, diversity, and community structure of bacterial communities associated with the fruit of S. carolinense are similar at fine spatial scales, suggesting there is no evidence of a distance-decay relationship. Thus, we are left with the question at what point do bacterial communities become distinct in the fruit of Solanum carolinense? In order to address this question, samples will need to be taken at larger spatial scales to determine when these communities become distinct. We also know that environment, pollinator, and host species play a large role in shaping microbial communities (Igwe & Vannette, 2019; Knief et al., 2010; Ottesen et al., 2016; Ushio et al., 2015; Wei & Ashman, 2018), but the degree to which these factors shape the bacterial communities found in Solanum carolinense fruit remains unclear. To address this, sampling over a large spatial scale should be done across a variety of environmental conditions. From that sampling, a comparison of similar environments, microbial communities will help to address the degree that environment plays a role in shaping microbial communities and determine if spatial scale plays a larger role than environment. To address the role of host species in shaping microbial communities, research should be conducted to compare the genotype of host species within a site to determine if in fact hosts that are closer related are more similar with regards to their microbial communities.

Supplemental Information

Supplemental Information 1 Sample information

Information on sample name, latitude, longitude, and number of samples collected at a site.

Click here for additional data file.

Supplemental Information 2 LNA and 16s rRNA V4 region primer sequences

Primer sequences used for locked nucleic acid PCR (LNA) and 16s rRNA PCR.

Click here for additional data file.

Supplemental Information 3 Sample information and reads input, filtered, denoised, tabled, and nonchim

Click here for additional data file.

Supplemental Information 4 Beta diversity (Jaccard Similarity) of sample locations

PCoA plot was used to determine differences between microbial communities associated with S. carolinense fruit and sample locations (PERMANOVA, p = 0.688, R2 = 0.30896). The axes indicate the percentage of variation in the data with axis 1 (the first component) representing 12.7% of the variation and axis 2 (the second component) representing 8% of the variation.

Click here for additional data file.

Supplemental Information 5 Statistical outputs

Statistical outputs for (A) observed richness, Faith’s phylogenetic diversity, (B) Jaccard index, (C) Bray–Curtis, and (D) Mantel.

Click here for additional data file.

We would also like to thank Andie Gonzales and Rafael Castañeda-Saldaña for helping with the field collections and Angie Estrada for helping with the LNA and Illumina PCR.

Additional Information and Declarations

Competing Interests

Author Contributions

Field Study Permissions

Data Availability

The authors declare there are no competing interests.

Ariel R. Heminger conceived and designed the experiments, performed the experiments, analyzed the data, prepared figures and/or tables, authored or reviewed drafts of the paper, and approved the final draft.

Lisa K. Belden, Jacob N. Barney, Brian D. Badgley conceived and designed the experiments, authored or reviewed drafts of the paper, and approved the final draft.

David C. Haak conceived and designed the experiments, analyzed the data, authored or reviewed drafts of the paper, and approved the final draft.

The following information was supplied relating to field study approvals (i.e., approving body and any reference numbers):

Field research experiments were approved by the Department of Parks and Recreation of town of Blacksburg, VA.

The following information was supplied regarding data availability:

The sequences are available at NCBI: PRJNA735185. The R code and the data is available at figshare: Haak, David; Heminger, Ariel (2021): Data: Horsenettle (Solanum carolinense) fruit bacterial communities are not variable across fine spatial scales. University Libraries, Virginia Tech. Dataset. https://doi.org/10.7294/14736411.v1.

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
