# Peer review of "Horsenettle (Solanum carolinense) fruit bacterial communities are not variable across fine spatial scales"

_PeerJ, doi:10.7717/peerj.12359_

## Round 0.1 · original submission · Major Revisions

Dear Author,

Please find attached the reviews from the Reviewers. Please update your manuscript accordingly. Good luck!

with kind regards,
Robert Czajkowski (Editor)

Reviewer 1 ·

Basic reporting

The article titled “Horsenettle (Solanum carolinense) fruit bacterial communities are not variable across fine spatial scales (#62008)” by Heminger AR et al., studied the diversity of microbial communities in fine special scales present in fruits of Horsenettle and presented that there are no significant differences were observed within the range they have tested.

Experimental design

The study design justifies the rationale of this work and the experiments were well designed but for comparison a distant spatial scale could have been added. As a reviewer from a reader’s point of view the below comments were made to improve the overall reach of this work.

Validity of the findings

Overall, the manuscript draft is well written and presented with good statistical analysis. The findings were of interest to the peers working around microbial diversity in plants related studies. There were few minor areas that need authors attention and needs clarifications. I have the below comments and concerns regarding the article.

Additional comments

Comments:
1. What is the effect of toxicity of Solinine glycoalkaloids present in the fruits of Solanum carolinense on the microbial communities? This is not discussed or considered in this study
2. Although the rationale for choosing Solanum carolinense was provided in lines 118-120. It is not adequate or justified. Please elaborate.
3. For comparison and to understand the microbial diversity, the authors could consider doing similar tests in distant spacial scales of Solanum carolinense or choose one more study site that is fine spatial scales. This would not only validate the observed results but also increases the significance of the study.
4. Line 19: Delete “that”
5. Line 143, 156, 157: Add a table for the primer sequences
6. Line 240: “significant variation” variation of which factor?
7. Line 270: correct to “fruit bacterial communities”
8. Line 321: Add space between 17 m.
9. Figure 1, 2 3: The authors have mentioned 9 sample locations. It is not clear from the map in Figure 1 as it shows only 8 locations.
10. As authors have mentioned in the conclusions: The sample number (n#) is very low ranging from 2-5. It is very difficult to attain any statistical significance due to this low numbers. I would highly encourage to increase the sample number as well as plan a similar fine spatial study for significance.
11. Figure 2: It is also “Enterobacterials” that make up high proportion of relative abundance. Why this is ignored in lines 227 and 228.
12. The authors can comment on Figure 2. Site locations 2,3, 8 and 9 about the Micrococales abundance.
13. Also, authors can write up briefly on other populations like Caulobacteriales, Betaptroteobacteriales and Flavibacterials abundance across different sites.
14. Figure 3: For site location 5: why are the values for this site location are very low? Any reason for the same?
15. Figure 4: A clustering is seen at -0.25 for site locations 2,8 and 3. This not discussed in the results. Rather no clustering was observed is stated (Line 245). Please explain.
16. In the discussion and in the Introduction the authors have stressed on the influence of these microbial communities in “host fitness” but this was not discussed in their results. Please discuss.
17. A limitations sections need to be added to this article. The points discussed in Discussion part like sample size etc need to be added.
18. The sample location is fine and not well separated (no effect of environment, soil pH, wind direction, pollinators, etc), All the limitations and the experiments to address them were already stated by the authors in Discussion and conclusions, in that case, what is the exact rationale of this study to investigate microbial diversity within closely distributed plant communities?

Reviewer 2 ·

Basic reporting

1, There is no figure legend for the supplemental figure, the authors should consider including it.

2, The sites and sample information should also be included in the supplemental table 1.

3, The format of “p value” should be consistent in the manuscript, p should be lowercase and italicized.

4, The authors used boxplot in figure 4 to show the distribution of observed Richness, Shannon, and Faith’s phylogenetic diversity by sampling locations. Considering the sample size is small, the authors should consider using barplots or just points to show the results.

5, Compared to the result section,the discussion section in the current draft is too long and confusing. The authors should write it succinctly and just cover the key finding of this study, which is mainly that distance decay in the bacterial communities associated with fruit is not observed at small scales.

6, The raw sequencing data hasn't been deposited on a public archive.

Experimental design

No comment

Validity of the findings

1, It seems that sample #73 (supplemental table 1 ) has very little input data, this might introduce extra bias for the analysis, maybe the authors should consider removing this sample if there are no other strong reasons to keep it.

Additional comments

The manuscript by Heminger et al., focuses on fruit house microbial communities in a perennial weed, Solanum carolinense. Previous studies from different labs were carried out mostly at large spatial scales. Compared to other tissues, the microbial microbiome in reproduction tissues are less studied. In the current study, the authors used 16S rRNA gene amplicon sequencing and studied the microbial communities in Solanum carolinense fruits at several different local sites within 0.5 km range. In contrast to the published observations in leaves, the bacterial community structure of these samples seems to be homogenous. There are no discernible differences observed regarding the richness or microbial community composition in this current study.

The results in this work challenged the "distance decay" in fruits at very fine scales. It's of interest to the field, if the author could address these points above, it will improve this manuscript.

---

## Round 0.2 · accepted · Accept

Dear Author, I am happy to inform you that your manuscript has been accepted for publication in PeerJ.
with kind regards,
Editor

Reviewer 2 ·

Basic reporting

No comments

Experimental design

No comments

Validity of the findings

No comments